DPM1 expression as a potential prognostic tumor marker in hepatocellular carcinoma

Li Ming 1
Xia Shengli victorxia@126.com 2
Shi Ping ship@ecust.edu.cn 1
1 State Key Laboratory of Bioreactor Engineering, East China University of Science and Technology , Shanghai , China
2 Department of Orthopedics, Shanghai University of Medicine & Health Sciences Affiliated Zhoupu Hospital , Shanghai , China
Nakai Kenta
Electronic publication date: 2020 Nov 24
Publication date: 2020
Volume: 8
Electronic Location ID: e10307
Received 2020 Jul 8; Accepted 2020 Oct 15
Copyright: ©2020 Li et al.
Copyright year: 2020
Copyright holder: Li et al.
License: This is an open access article distributed under the terms of the Creative Commons Attribution License, which permits unrestricted use, distribution, reproduction and adaptation in any medium and for any purpose provided that it is properly attributed. For attribution, the original author(s), title, publication source (PeerJ) and either DOI or URL of the article must be cited.
License URL: https://creativecommons.org/licenses/by/4.0/

Keywords: DPMS, Liver cancer, Biomarker, Bioinformatics analysis, Prognostic value

Funding: National Natural Science Foundation of China 31671309 Training Planned Fund of Academic Leaders, Shanghai Pudong New Area Health System PWR 12018-09 This work was sponsored by grants from the National Natural Science Foundation of China (31671309), and the Training Planned Fund of Academic Leaders, Shanghai Pudong New Area Health System (PWR 12018-09). The funders had no role in study design, data collection and analysis, decision to publish, or preparation of the manuscript.

==============================
Background

Altered glycosylation of proteins contributes to tumor progression. Dolichol phosphate mannose synthase (DPMS), an essential mannosyltransferase, plays a central role in post-translational modification of proteins, including N-linked glycoproteins, O-mannosylation, C-mannosylation and glycosylphosphatidylinositol anchors synthesis. Little is known about the function of DPMS in liver cancer.

Methods

The study explored the roles of DPMS in the prognosis of hepatocellular carcinoma using UALCAN, Human Protein Atlas, GEPIA, cBioPortal and Metascape databases. The mRNA expressions of DPM1/2/3 also were detected by quantitative real-time PCR experiments in vitro.

Results

The transcriptional and proteinic expressions of DPM1/2/3 were both over-expressed in patients with hepatocellular carcinoma. Over-expressions of DPMS were discovered to be dramatically associated with clinical cancer stages and pathological tumor grades in hepatocellular carcinoma patients. In addition, higher mRNA expressions of DPM1/2/3 were found to be significantly related to shorter overall survival in liver cancer patients. Futhermore, high genetic alteration rate of DPMS (41%) was also observed in patients with liver cancer, and genetic alteration in DPMS was associated with shorter overall survival in hepatocellular carcinoma patients. We also performed quantitative real-time PCR experiments in human normal hepatocytes and hepatoma cells to verify the expressions of DPM1/2/3 and results showed that the expression of DPM1 was significantly increased in hepatoma cells SMMC-7721 and HepG2.

Conclusions

Taken together, these results suggested that DPM1 could be a potential prognostic biomarker for survivals of hepatocellular carcinoma patients.

Introduction

Hepatocellular carcinoma (HCC) is one of the most frequently and commonly occurring malignant tumors worldwide. The global incidence and mortality rate of HCC are ranked 5th and 3rd among all types of cancers (Jemal et al., 2011; Siegel et al., 2014). Despite making remarkable advances in new technologies for diagnosis and treatment, the incidence and mortality of HCC still continue to growth because of the poorest prognosis (Llovet, Burroughs & Bruix, 2003; Maluccio & Covey, 2012). Therefore, it is urgently needed to determine reliable predictive biomarkers for early diagnosis and accurate prognosis, and to develop new molecular targeted therapeutic strategies.

The occurrence and development of several cancer types are closely associated with aberrant protein glycosylation (Pinho & Reis, 2015; Stowell, Ju & Cummings, 2015). Studies have suggested that altered glycosylation of proteins has been observed in liver cancer (Mehta, Herrera & Block, 2015). Although mounting evidence has reported the role of glycosylation in tumor progression (Hakomori, 2002; Fuster & Esko, 2005; Reis et al., 2010), there is limited information on how glycosylation affects the liver cancer development. Recent studies have focused on glycosylation crosstalks with cellular metabolism and related kinases (Butt et al., 2012; Itkonen & Mills, 2013; Wang et al., 2016; Nguyen et al., 2017).

Dolichol phosphate mannose synthase (DPMS), an essential mannosyltransferase, plays a central role in post-translational modification of proteins, including N-linked glycoproteins, O-mannosylation, C-mannosylation and glycosylphosphatidylinositol (GPI) of proteins (Maeda & Kinoshita, 2008). It has three subunits containing DPM1, DPM2 and DPM3 in human. DPM1, a mainly catalytic component of DPMS, is composed of 260 amino acids without any transmembrane domain region (Colussi, Taron & Mack, 1997; Tomita et al., 1998). DPM2 and DPM3 are regulatory subunits that help DPM1 localize on the endoplasmic reticulum membrane and enable it to exert catalytic activity (Maeda et al., 2000). The most reported finding on the DPMS gene is that its absence activity is associated with congenital diseases of glycosylation (CDG) and a defect in DPM1 has been identified to cause CDG-Ie (Kim et al., 2000; Imbach et al., 2000). In addition to this, studies have reported that abnormal expression or altered enzymatic activity of DPMS was related to cell proliferation and angiogenesis. Increased DPMS activity in bovine capillary endothelial cells correlated with rised cellular proliferation (Baksi et al., 2009). Moreover, previous studies also reported that overexpressing DPMS in capillary endothelial cells significantly enhanced angiogenesis and strengthened wound healing (Zhang et al., 2010). DPMS activity, however, was lacking and subsquently led to cell cycle arrest and induction of apoptosis in tunicamycin-treated capillary endothelial cells (Banerjee, 2012). Reduced gene expression of DPMS also decreased the cellular angiogenic potential (Baksi et al., 2016). These research results indicate that the genes encoding DPMS and its protein activity may be positively related to tumor progression. However, the specific role of DPMS remains unclear in the development and progression of liver cancer. In this present work, we solved this problem by analyzing the expressions and genetic alterations of three subunits of DPMS and their association with clinical parameters in HCC patients. Furthermore, we also analyzed the predicted functions and pathways of DPMS as well as their similar genes.

Material and Methods

Datasets

Datasets used for correlation analysis between DPM1/2/3 and chronic liver disease (CLD) were obtained from GEO database (http://www.ncbi.nlm.nih.gov/geo/) after searching for keywords related to CLD. We selected three separate gene expression profiles (GSE114783, GSE128726 and GSE89632) for our study and the detailed information of the datasets was shown in Table 1. The data used for ROC curve plotted were collected from TCGA LIHC datasets. The figures about the relationship between DPM1/2/3 expression and CLD were drawn using R package, ggplot2 v3.3.2. The significance of DPM1/2/3 expressions between normal and CLD samples were analyzed via unpaired Student’s t-test. The ROC curves were created by R package, pROC v1.16.2.

Table 1 Detailed information of the GEO datasets in this study.

Datasets ID	Species	Data type	Platform	Disease type	Normal number	Patient number	
GSE114783	Homo sapiens	Expression profiling by array	GPL15491	HBV-related liver cirrhosis	3	10	
GSE128726	Homo sapiens	Expression profiling by array	GPL21185	HCV-related liver cirrhosis	9	10	
GSE89632	Homo sapiens	Expression profiling by array	GPL14951	Non-alcoholic steatohepatitis	24	19	

UALCAN

UALCAN (http://ualcan.path.uab.edu) is a comprehensive, user-friendly, and interactive web resource and provides data online analysis and mining based on cancer OMICS data (TCGA and MET500). It is designed to analyze relative transcriptional expression of potential genes of interest between tumor and normal samples and association of the transcriptional expression with relative clinicopathologic parameters. In addition, it is also used to evaluate epigenetic regulation of gene expression and pan-cancer gene expression (Chandrashekar et al., 2017). In our study, UALCAN was used to analyze the mRNA expressions of three subunits of DPMS in HCC samples and their relationship with clinicopathologic parameters. Difference of transcriptional expression or pathological stage analysis was compared by Student’s t-test and p <0.05 was considered as statically significant.

Human Protein Atlas

The Human Protein Atlas (https://www.proteinatlas.org) is a website that provides human proteins data in cells, tissues and organs, including immunohistochemistry-based expression data for near 20 common kinds of cancers (Asplund et al., 2012). The database can be conveniently used to compare the protein differential expressions of interest genes in tumors and normal tissues. In this study, direct comparison of protein expression of three subunits of DPMS between human normal and HCC tissues was performed by immunohistochemistry image.

GEPIA

Gene Expression Profiling Interactive Analysis (GEPIA) is a database developed and built by the team of professor Zhang of Peking University based on the data of the UCSC Xena project. It is an interactive web server that can dynamically analyze and visualize TCGA (The Cancer Genome Atlas) gene expression profile data. It can provide customizable and powerful functions, including differential expression analysis between tumor and normal samples, profiling plotting, survival analysis, similar gene detection, and so on Tang et al. (2017). In the current study, we operated correlative prognostic analysis and similar gene detection of DPM1, DPM2 and DPM3, respectively. p < 0.05 was considered as statically significant. The significance of expression analysis was completed using Student’s t-test. Kaplan–Meier curve was used to accomplish prognostic analysis.

cBioPortal

cBioPortal (http://www.cbioportal.org), an online open-access website resource, can display multidimensional cancer genomics data in a visual form. It can also help researchers explore the genetic changes between samples, genes and pathways, and combine them with clinical results (Gao et al., 2013). In this experiment, we studied the genomic profiles of DPMS three subunits, which included putative copy-number alterations (CNAs) from genomic identification of significant targets in cancer (GISTIC) and mRNA Expression z-Scores (RNASeq V2 RSEM) were gained with a z-score threshold ±1.8. Genetic alterations in DPMS and their association with overall survival (OS) and disease free survival (DFS) of HCC patients were exhibited as Kaplan–Meier plots and log-rank test was implemented to confirm the significance of the difference between the survival curves, and when a p value <0.05, the difference was statically significant.

Metascape

Metascape (http://metascape.org), a free and credible gene-list analysis device, can be used for gene annotation analysis and function analysis. It is a mechanized meta-analysis device that can realize habitual and different pathways in a set of orthogonal target-discovery studies (Zhou et al., 2019). In this work, Metascape was used to implement function and pathway enrichment analysis of DPMS members and their similar genes that acquired using GEPIA. Statistically significant difference was p < 0.05 and minimum enrichment number was 3. Databases containing OmniPath and BioGrid were used for protein-protein interactions enriched analysis. Futhermore, Molecular Complex Detection (MCODE) was supposed to recognize closely related protein components.

Cell culture

The human hepatoma cells SMMC-7721, HepG2 and immortal hepatic cell QSG-7701 involved in the experiment were gained from Institute of Cell Biology (Shanghai, China). All cell lines were cultured in RPMI-1640 or DMEM medium (Gibco/Invitrogen, Camarillo, CA, UNITED STATES) supplied with 10% fetal bovine serum (PAN-Biotech, Aidenbach, Germany), and then all cells were incubated at 37 °C in a 5% CO2 environment.

RT-qPCR

TRIeasy™ Total RNA Extraction Reagent (Yeasen, Shanghai, China) was used for total RNA extraction, and then the total RNA was reverse transcribed to cDNA with the Hifair® 1st Strand cDNA Synthesis Kit (Yeasen, Shanghai, China) according to the product instruction. Hieff UNICON®Power qPCR SYBR Green Master Mix (Yeasen, Shanghai, China) was used to conduct RT-qPCR experiment on a Bio-Rad CFX96 System (Bio-Rad, Hercules, CA, USA). The reaction conditions were as follows: pre-denaturation at 95 °C for 30 s, followed by 40 cycles of amplification at 95 °C for 10 s and 60 °C for 30 s. Relative mRNA expression levels of DPM1/2/3 were measured based on the 2−ΔΔCt method with 18S used for normalization. The significance of expression analysis was completed using Student’s t-test. Table 2 showed the primers we used in this study.

Table 2 Primers used for quantitative real-time PCR.

Gene	Primers	Sequences (5′→3)	
DPM1	Forward
Reverse	ACAGGAAGTTTCAGATTATACCGAA ATTCACCATAAACACGATCCACA	
DPM2	Forward
Reverse	GCATCCTTAGCCGCTACACT
GCGTTTGCCATGCCTAAGAG	
DPM3	Forward
Reverse	TCGCAGTGACCATGACGAAA
TTAGGCTGTCAGAAGCGCAG	
18S	Forward
Reverse	CGGCTACCACATCCAAGGAAG
AGCTGGAATTACCGCGGCT	

Results

Transcriptional levels of DPMS in liver cancer

In order to explore the gene expressions of three subunits of DPMS in different types of cancer, mRNA expressions of DPM1, DPM2 and DPM3 were analyzed by UALCAN. As was shown in Fig. 1, we observed that DPM1, DPM2 and DPM3 had higher mRNA expressions for most kinds of tumor samples compared to normal samples, respectively. For example, mRNA expression levels of DPM1 and DPM2 were very highly expressed in colon adenocarcinoma (COAD) (DPM1, p = 1.62E−12; DPM2, p <1E−12), head and neck squamous cell carcinoma (HNSC) (DPM1, p <1E−12; DPM2, p = 1.62E−12), esophageal carcinoma (ESCA) (DPM1, p = 1.22E−07; DPM2, p = 2.30E−02), liver hepatocellular carcinoma (LIHC) (DPM1, p = 1.62E−12; DPM2, p <1E-12), rectum adenocarcinoma (READ) (DPM1, p = 4.07E−09; DPM2, p = 1.62E−12 ) and so on (Figs. 1A, 1B). Similarly, DPM3 gene was particularly highly expressed in breast invasive carcinoma (BRCA) (p = 1.62E−12), ESCA (p = 8.22E−10), LIHC (p = 1.11E−16) and glioblastoma multiforme (GBM) (p = 1.53E−05) (Fig. 1C). Thus, our results showed that transcriptional expressions of DPMS were significantly over–expressed in many different types of cancer. In particular, all three subunits of DPMS were expressed highly in LIHC and ESCA. Next, we examined the specific mRNA expressions of DPM1, DPM2 and DPM3 in liver tumor using UALCAN database. As was shown in Figs. 2A, 2B and 2C, mRNA expressions of three genes were all found significantly up-regulated in HCC tissues compared to normal samples (all p < 0.001). We next performed the protein expression levels of DPMS in HCC using Human Protein Atlas database. Results indicated that medium and low protein expressions of DPM1 and DPM3 were expressed in normal liver tissues, while high protein expressions of them were showed in HCC tissues (Figs. 2D, 2F). In addition, DPM2 protein were not detected in normal liver tissues, whereas medium expression of DPM2 were observed in HCC tissues (Fig. 2E). In general, the results indicated that transcriptional and proteinic expressions of DPMS were both over-expressed in patients with HCC.

Figure 1 Transcriptional expressions of (A) DPM1, (B) DPM2 and (C) DPM3 in different types of cancer diseases (UALCAN database).

Blue: Normal; Red: Tumor. Abbreviations: BLCA, Bladder urothelial carcinoma; BRCA, Breast invasive carcinoma; CESC, Cervical squamous cell carcinoma; CHOL, Cholangiocarcinoma; COAD, Colon adenocarcinoma; ESCA, Esophageal carcinoma; GBM, Glioblastoma multiforme; HNSC, Head and Neck squamous cell carcinoma; KICH, Kidney chromophobe; KIRC, Kidney renal clear cell carcinoma; KIRP, Kidney renal papillary cell carcinoma; LIHC, Liver hepatocellular carcinoma; LUAD, Lung adenocarcinoma; LUSC, Lung squamous cell carcinoma; PAAD, Pancreatic adenocarcinoma; PRAD, Prostate adenocarcinoma; PCPG, Pheochromocytoma and Paraganglioma; READ, Rectum adenocarcinoma; SARC, Sarcoma; SKCM, Skin cutaneous melanoma; THCA, Thyroid carcinoma; THYM, Thymoma; STAD, Stomach adenocarcinomna; UCEC, Uterine corpus endometrial carcinoma.

Figure 2 The mRNA and protein expressions of DPMS in HCC and normal liver tissues.

(A–C) mRNA expressions of DPM1, DPM2 and DPM3 in HCC tissues compared to normal samples (UALCAN database). ∗∗∗p < 0.001. (D–F) Representative immunohistochemistry images of DPM1, DPM2 and DPM3 in HCC tissues and normal liver tissues (Human Protein Atlas).

Figure 3 Association of mRNA expressions of DPMS with tumor grades and patients’ individual cancer stages in HCC patients (UALCAN).

(A–C) Association of mRNA expressions of DPM1, DPM2 and DPM3 with tumor grades in HCC patients. (D–F) Relationship between mRNA expressions of DPM1, DPM2 and DPM3 and individual cancer stages of HCC patients. ∗p < 0.05, ∗∗p < 0.01, ∗∗∗p < 0.001.

Relationship between the mRNA levels of DPMS and the clinicopathological parameters in liver cancer patients

Because we observed mRNA and protein levels of DPMS were over-expressed in HCC patients, we subsequently investigated the connection between mRNA expressions of DPMS members with clinicopathological features of HCC patients with UALCAN, containing tumor grades and patients’ individual cancer stages. As presented in Fig. 3, mRNA expressions of DPMS members were significantly associated with tumor grades, and the mRNA expressions of DPMS headed to be higher with tumor grade elevated. The maximum mRNA expressions of DPM1/2 were showed in tumor grade 4 (Figs. 3A, 3B), whereas the supreme mRNA expression of DPM3 was found in tumor grade 3 (Fig. 3C). The reason why mRNA expression of DPM3 in grade 3 seemed to be higher than that in grade 4 may be attributed to the small sample size (only 12 HCC patients at grade 4). Similarly, the mRNA expressions of DPMS were noticeably related to the cancer stage of patients so, the patients with more advanced cancer, the higher in mRNA expressions of DPMS. The highest mRNA expressions of DPM1/2 were observed in tumor stage 3 (Figs. 3D, 3E), while the maximum DPM3 mRNA expression was noticed in stage 4 (Fig. 3F). Briefly, the results above indicated that mRNA expressions of DPMS were obviously associated with pathological parameters in HCC patients. Moreover, HCC usually developed from CLD caused by hepatitis B virus (HBV) and hepatitis C virus (HCV) infection, fatty liver and so on. The relationships between the expressions of DPM1/2/3 and CLD including HBV-related liver cirrhosis, HCV-related liver cirrhosis and non-alcoholic steatohepatitis were also analyzed. Three suitable datasets (GSE114783, GSE128726 and GSE89632) were chosen for verifying the expression of DPM1/2/3 in CLD. We found that the expressions of DPM1/2/3 in HBV and HCV-related liver cirrhosis and non-alcoholic steatohepatitis samples were more or less higher than normal samples (Fig. 4). Therefore, the expressions of DPM1/2/3 were also related to disease development of HCC.

Prognostic value of mRNA expression of DPMS in liver cancer patients

To assess the value of differentially expressed DPMS in the progression of HCC, we used GEPIA to evaluate the relationship between differentially expressed DPMS and clinical outcome. OS curves were presented in Fig. 5. We detected that liver cancer patients with low transcriptional levels of DPM1 (p = 0.007), DPM2 (p = 0.0032) and DPM3 (p = 0.029), were significantly connected with longer OS (Figs. 5A, 5B, and 5C). The worth of differentially expressed DPMS in the DFS of HCC patients was also estimated. Noteworthy, the longer DFS indicated to the HCC patients with lower DPM2 transcriptional levels (p = 0.049) (Fig. 5E). The receiver operating characteristic (ROC) curves were used to detect the prediction accuracy of DPM1/2/3 in distinguishing the HCC from the normal samples compared with existed liver tumor markers containing alpha-fetoprotein (AFP), glypican-3 (GPC-3) and transforming growth factor- β1 (TGFβ1). Our results indicated that DPM1/2/3 had a better performance than AFP and TGFβ1 for the diagnosis of HCC (Fig. 6). Area under the curve (AUC) of the DPM1/2/3 were 0.709, 0.860 and 0.746, respectively (Figs. 6A, 6B and 6C). AUC of the existed tumor markers including AFP, GPC-3 and TGFβ1 were 0.679, 0.879 and 0.577, respectively (Figs. 6D, 6E and 6F). Taken together, DPM1/2/3 may be also potential biomarkers for diagnosis or screening of HCC besides AFP, GPC-3 and TGFβ1.

Figure 4 The mRNA expressions of DPM1/2/3 in normal liver samples and CLD samples.

(A–C) The expressions of DPM1/2/3 in normal and HBV-related liver cirrhosis. (D–F) The expressions of DPM1/2/3 in normal and HCV-related liver cirrhosis. (G–I) The expressions of DPM1/2/3 in normal and non-alcoholic steatohepatitis. Abbreviations: HBV-LC, HBV-related liver cirrhosis; HCV-LC, HCV-related liver cirrhosis; NASH, non-alcoholic steatohepatitis.

Figure 5 The prognostic value of different expressed DPM1, DPM2 and DPM3 in HCC patients (GEPIA).

(A–C) Overall survival curves of DPM1, DPM2 and DPM3. (D–F) Disease free survival curves of DPM1, DPM2 and DPM3.

DPMS genetic alteration and similar gene network in patients with HCC

Next, we implemented a universal analysis of the molecular characteristics of differentially expressed DPMS. Genetic variations of differentially expressed DPMS in HCC was analyzed utilizing cBioPortal. A total of 366 samples from TCGA pan cancer database were studied, and altered gene set or pathway was detected in 151queried samples (alteration rate was 41%). The alteration rates of DPM1, DPM2, and DPM3 were 19%, 6% and 24%, respectively (Figs. 7A, 7B). The most prevalent change in these samples was enhanced mRNA expression. The Kaplan–Meier plotter results and log-rank test presented a considerable difference in OS (p = 0.0264), but no remarkable difference in DFS (p = 0.0841) between the samples with changes in one of the target genes and those without variations in any target genes (Figs. 7C, 7D).

Functional enrichment analysis of DPMS in patients with HCC

Top 50 genes similar to DPM1, DPM2 and DPM3 respectively (a total of 150 genes) were searched by GEPIA (Table S1). Next, the functions of DPMS and their similar genes were predicted by analyzing GO and KEGG in Metascape. The top 20 GO enrichment items were classified into three functional groups: biological process group, molecular function group, and cellular component group (Figs. 8A, 8B and Table 3). The DPMS members and their similar genes were mainly enriched in biological processes such as ncRNA processing, DNA repair, viral gene expression, deoxyribonucleoside triphosphate metabolic process and so on. The molecular functions regulated by DPMS and their similar genes were snRNP binding, ubiquitin binding, nucleotidyltransferase activity and ubiquitin-like protein transferase activity. The cellular components affected by DPMS and their similar genes were involved in transferase complex, methyltransferase complex, chromosomal region and nucleolar part.

Figure 6 The assessment of the diagnosis effect among DPM1/2/3 and existing markers in normal and HCC using the ROC curve.

(A–C) ROC curves and AUC values of DPM1, DPM2 and DPM3 respectively. (D–F) ROC curves and AUC values of AFP, GPC3 and TGFβ1 respectively.

Figure 7 Genetic alterations in DPMS and their association with OS and DFS in HCC patients (cBioPortal).

(A) Summary of alterations in DPMS. (B) OncoPrint visual summary of alteration on a query of DPMS. (C) Kaplan–Meier plots comparing OS in cases with/without DPMS gene alterations. (D) Kaplan–Meier plots comparing DFS in cases with/without DPMS gene alterations.

Figure 8 The enrichment analysis of DPMS and their similar genes in HCC (Metascape).

(A) Heatmap of Gene Ontology (GO) enriched terms colored by p-values. (B) Network of GO enriched terms colored by p-value, where terms containing more genes tend to have a more significant p-value. (C) Heatmap of Kyoto Encyclopedia of Genes and Genomes (KEGG) enriched terms colored by p-values. (D) Network of KEGG enriched terms colored by p-value, where terms containing more genes tend to have a more significant p-value. (E) Protein–protein interaction (PPI) network and three most significant MCODE components form the PPI network. (F) Independent functional enrichment analysis of three MCODE components.

The 6 most significant KEGG pathways for the DPMS and their similar genes were displayed in Figs. 8C, 8D and Table 4. These pathways comprised pyrimidine metabolism, RNA transport, ubiquitin mediated proteolysis, mTOR signaling pathway and so on. Moreover, for more comprehending the relationship between DPMS and HCC, we performed enrichment analysis of protein–protein interaction with Metascape. Figures 8E and 8F exhibited the protein interaction correlation and important MCODE components. The top 3 essential MCODE components were achieved from the protein–protein interaction network. After function and pathway enrichment analysis for each MCODE constituents respectively, the results demonstrated that biological functions regulated by DPMS and their similar genes were mainly related to mRNA and RNA splicing, protein export form nucleus and nucleocytoplasmic transport.

The mRNA expression levels of DPM1/2/3 in vitro

We evaluated DPM1, DPM2 and DPM3 expression levels in a panel of three cell lines: two hepatoma cells (HepG2 and SMMC-7721) and one normal liver cell line (QSG-7701). The mRNA expression measured by RT-qPCR revealed that DPM1 transcription levels in cancerous cell lines were higher than that in normal liver cells (Fig. 9A) and the result was consistent with our prediction. Moreover, the expression of DPM2 and DPM3 in SMMC-7721 cell was significantly increased, while those expression did not change significantly in HepG2 cell (Figs. 9B, 9C). This discrepancy may be due to a number of differences between cell types and more cell and tissue samples are needed to validate the results. Therefore, DPM1 could be the most potential prognostic biomarker for survivals of HCC patients.

Discussion

Abnormal glycosylation has been found in human cancer cells decades ago, and more and more researchers have discovered that protein glycosylation contributed to tumor metastasis, angiogenesis and progression (Oliveira-Ferrer, Legler & Milde-Langosch, 2017; Cheng & Oon, 2018). Being an essential component of glycosyltransferase complex, DPMS protein is involved in multiple protein glycosylation process, including N-glycosylation, O-glycosylation, C-mannosylation and GPI anchors synthesis (Maeda & Kinoshita, 2008). Many studies have reported that overexpressed DPMS promoted cell proliferation and angiogenesis (Zhang et al., 2010), and silencing DPMS with shRNA significantly reduced cell growth (Baksi et al., 2016). Moreover, increased DPMS activity also accelerated cellular growth (Baksi et al., 2009; Banerjee, 2012). In view of the above results, we speculated that DPMS may be related to tumorigenesis and progression. To confirm this hypothesis, we predicted the expression of DPMS in cancer through bioinformatics methods, especially in liver cancer. In addition, genetic alteration and prognostic values of three subunits of DPMS in HCC were also analyzed.

Table 3 The GO function enrichment analysis of DPM1/2/3 and their similar genes in HCC.

GO	Category	Description	Count	%	Log10(P)	Log10(q)	
GO:0034470	GO Biological Processes	ncRNA processing	11	9.32	−5.54	−2.25	
GO:0006281	GO Biological Processes	DNA repair	13	11.02	−5.42	−2.21	
GO:0019080	GO Biological Processes	viral gene expression	7	5.93	−4.32	−1.25	
GO:0009200	GO Biological Processes	deoxyribonucleoside triphosphate metabolic process	3	2.54	−4.06	−1.03	
GO:0000726	GO Biological Processes	non-recombinational repair	5	4.24	−3.74	−0.8	
GO:0071900	GO Biological Processes	regulation of protein serine/threonine kinase activity	10	8.47	−3.57	−0.79	
GO:0008033	GO Biological Processes	tRNA processing	5	4.24	−3.41	−0.74	
GO:0032006	GO Biological Processes	regulation of TOR signaling	4	3.39	−2.81	−0.38	
GO:0072594	GO Biological Processes	establishment of protein localization to organelle	8	6.78	−2.26	−0.1	
GO:0006353	GO Biological Processes	DNA-templated transcription, termination	3	2.54	−2.23	−0.09	
GO:0006412	GO Biological Processes	translation	9	7.63	−2.04	0	
GO:1990234	GO Cellular Components	transferase complex	18	15.25	−7.42	−3.54	
GO:0034708	GO Cellular Components	methyltransferase complex	6	5.08	−4.89	−1.74	
GO:0061695	GO Cellular Components	transferase complex, transferring phosphorus-containing groups	7	5.93	−3.6	−0.79	
GO:0098687	GO Cellular Components	chromosomal region	6	5.08	−2.11	−0.04	
GO:0044452	GO Cellular Components	nucleolar part	4	3.39	−1.96	0	
GO:0070990	GO Molecular Functions	snRNP binding	4	3.39	−7.73	−3.54	
GO:0043130	GO Molecular Functions	ubiquitin binding	4	3.39	−3.19	−0.63	
GO:0016779	GO Molecular Functions	nucleotidyltransferase activity	4	3.39	−2.47	−0.25	
GO:0019787	GO Molecular Functions	ubiquitin-like protein transferase activity	7	5.93	−2.4	−0.2	

Results from our study showed that the transcriptional levels of DPMS were highly expressed in different types of cancer. Moreover, over-expressions of mRNA and protein were both found in three subunits of DPMS, and mRNA expressions of DPMS were significantly associated with patients’ individual cancer stages and tumor grades in HCC patients. Besides, higher mRNA expressions of DPM1/2/3 were significantly associated with shorter OS in liver cancers patients. Meanwhile, higher mRNA expression of DPM2 was significantly associated with shorter DFS in liver cancers samples. These data demonstrated that differentially expressed DPMS may play a significant role in HCC. Since three subunits of DPMS were significantly differentially expressed in HCC and closely related to liver tumor prognosis, we next explored their molecular characteristics in HCC. High alteration rate (41%) of DPMS was observed in HCC patients and the genetic alteration in DPMS was associated with shorter OS in HCC patients. Tumorigenesis and development of HCC is sophisticated and various, and genetic alteration exerts an important function among this process (Yap et al., 2015). Among the genetic alteration elevated mRNA expression and gene amplification were the most common changes. Gene amplification, or genomic DNA copy number aberration, is frequently observed in some solid tumors and has been thought to contribute to tumor evolution (Klein & Klein, 1986; Albertson et al., 2003; Albertson, 2006). Therefore, the high alteration of gene amplification in DPMS may be related to liver cancer progression. However, the specific function of gene amplification of DPMS in liver cancer need to be further studied. Finally, functions and pathways of DPM1/2/3 and their total 150 similar genes in HCC patients were analyzed. Biological processes such as ncRNA processing and DNA repair, cellular components such as transferase complex, molecular functions snRNP binding and ubiquitin binding, signal pathways such as RNA transport were remarkably regulated by DPMS and their similar genes in HCC. Our findings that DPMS was highly expressed in tumor cells are consistent with the conclusion that overexpression of DPMS in capillary endothelial cells promoted cell proliferation (Zhang et al., 2010). In addition, a paper noted that upregulation of DPMS activity may involve in angiogenesis for breast and other solid tumor proliferation and metastasis and identified DPMS as a potential “angiogenic switch” (Baksi et al., 2009). Another report related to prostate tumor invasion pointed out DPM3 was a invasion suppressor using microarray expression analysis of the transcription levels in prostate cancer sublines (Manos et al., 2001). This result is inconsistent with our conclusion that DPM3 was over-expressed in liver cancer cells, and the relationship between DPM3 and the invasion ability in liver cancer cells is worth further study. In addition to the above, the abnormal expressions of DPMS have been reported to be associated with human health, such as aging (Kousvelari et al., 1988), Thy-1 lymphoma (Nozaki et al., 1999) and CDG (Kim et al., 2000; Haeuptle & Hennet, 2009). These findings may help us to deepen our understanding for the role of DPMS in tumorigenesis and specific action mechanism among cancers.

Table 4 The KEGG function enrichment analysis of DPMS and their similar genes in HCC.

GO	Category	Description	Count	%	Log10(P)	Log10(q)	
hsa03040	KEGG Pathway	Spliceosome	6	5.08	−4.29	−1.59	
hsa00240	KEGG Pathway	Pyrimidine metabolism	5	4.24	−3.8	−1.57	
hsa03013	KEGG Pathway	RNA transport	6	5.08	−3.7	−1.57	
hsa04120	KEGG Pathway	Ubiquitin mediated proteolysis	4	3.39	−2.34	−0.55	
hsa05100	KEGG Pathway	Bacterial invasion of epithelial cells	3	2.54	−2.21	−0.47	
hsa04150	KEGG Pathway	mTOR signaling pathway	3	2.54	−1.42	0	

Figure 9 The mRNA expression levels of (A) DPM1, (B) DPM2 and (C) DPM3 in normal liver cells and hepatoma cell lines.

∗p < 0.05, ∗∗∗p < 0.001.

It is known that HCC generally occurs in patients withCLD as a result of HBV and HCV infections, nonalcoholic fatty liver disease and alcohol-use disorder (Villanueva, 2019). The occurrence of CLD caused by above factors is related to the glycosylation changes of key proteins (Sawamura et al., 1984; Burgess, Baenziger & Brown, 1992; Ihara et al., 1998; Ono & Hakomori, 2003; Lee et al., 2004). For example, hepatocytes in transgenic mice that specifically expressed N-acetylglucosaminyltransferase III (GnT-III) had a swollen oval-like morphology and many lipid droplets (Ihara et al., 1998; Lee et al., 2004). GnT-III was also likely to play essential roles in the change of glycosylation in viral infected people with liver diseases. DPMS is upstream of GnT-III and whether DPMS participates in the regulation process of this enzyme is worth further studying. In addition, ethanol oxidation products such as acetaldehyde interfered with the N-glycan biosynthesis and/or transfer by binding the involved enzymes in patients with liver disease. Modified glycosylation influenced proteins and receptors binding of the sinusoidal and cell surfaces of the liver in diverse CLD. Main membrane receptors glycosylation orchestrated their function in controlling tumor cell adhesion, motility and invasiveness (Ono & Hakomori, 2003). Furthermore, modification in glycosylated receptor assignment and concentration led to glycoproteins accumulation, which were associated with the tumor size in HCC patients (Sawamura et al., 1984; Burgess, Baenziger & Brown, 1992). Hence, the etiology of liver cancer due to chronic liver disease is perhaps attributed to the major membrane receptors and DPMS as an essential mannosyltransferase may be involved in glycosylation of major membrane receptors in liver cancer.

Meanwhile, alterations in glycosylation are a common feature of cancer cells, and the complexity in protein glycosylation improves cell molecules functional diversity (Clerc et al., 2016). Many glycosyltransferases such as N-acetylglucosaminyltransferase V (GnT-V), N-acetylglu-cosaminyltransferase III (GnT-III) and α1-6 fucosyltransferase (FUT8) have been considered to be related to the development of HCC. Genomic analysis of HCC patients inspired that overexpressed of FUT8 gene, the cause of core fucosylation, indicated that these glycan changes promoted hepatocarcinogenis, letting them potential tumor biomarkers and therapeutic targets (Cancer Genome Atlas Research Network, 2017). Studies have shown that expression changes of fucosyltransferase 1 and β-1,3-galactosyltransferase 5 led to the occurrence of HCC (Kuo et al., 2017). High expression of these enzymes in liver cancer patients was closely linked to shorter survival times of HCC patients (Sun et al., 1995). DPMS is upstream of these enzymes and the expression of DPMS is closely related to the expression of these enzymes. Therefore, DPMS may influence prognosis of HCC via affecting these related enzymes or similar mechanisms with these enzymes.

So far AFP, des- γ-carboxy-prothrombin (DCP) ,GPC3 and TGFβ1 are the major already-existed cancer biomarkers for HCC (Tateishi et al., 2008). These biomarkers could be used for early detection of HCC and as markers of recurrence in the follow-up of HCC patients. AFP is more sensitive to the diagnosis of HCC, but its specificity is lower than that of DCP (Marrero et al., 2003; Marrero et al., 2009). Soluble GPC3 is more sensitive than AFP in monitoring highly or moderately differentiated HCC. Simultaneous detection of two or more markers increases the overall sensitivity from 50% to 72% (Hippo et al., 2004). However, about 30% of HCC patients are still negative for these traditional tumor markers. In our study, DPM1 could be a potential prognostic biomarker for survivals of HCC patients. Therefore, it is possible to use DPM1 as an effective supplemental biomarker of liver cancer. The combined application of DPM1 and other already-existed biomarkers would greatly improve the early diagnosis and accurate prognosis of liver cancers. Our study also has some limitations. First, despite mRNA expressions of DPM1/2/3 were related to the prognosis of HCC, all the data performed in our research were obtained from the online website, further studies containing larger sample sizes are needed to confirm our results and to explore the clinical application of the DPMS in HCC treatment. Second, we did not assess the potential diagnostic and therapeutic roles of DPMS in HCC, so future studies are required to explore whether DPMS could be used as diagnostic markers or as therapeutic targets. Finally, we did not explore the potential mechanisms of DPMS in HCC. Future studies are worth to investigate the detailed mechanism between DPMS expression and HCC.

Conclusion

In this paper, we studied the expressions of DPM1/2/3 in tumor cells and its relationship with tumorigenesis for the first time. Our results showed that over-expressions of DPM1/2/3 were significantly associated with clinical cancer stages and pathological tumor grades in HCC patients. Besides, higher mRNA expressions of DPM1/2/3 were found to be significantly connected with OS in HCC patients. Moreover, high genetic alteration rate of DPM1/2/3 (41%) was also observed, and genetic alteration in DPM1/2/3 was associated with shorter OS in HCC patients, which provide a better understanding of molecular targets for improved liver cancer therapeutic strategies in the future. DPM1 was the most potential prognostic biomarker for liver cancer via cell experiment verified. To sum up, these results indicated that DPM1 could be a prognostic biomarker for survivals of HCC patients.

Supplemental Information

Supplemental Information 1 Similar genes for DPM1/2/3

Click here for additional data file.

Supplemental Information 2 Raw data for RT-qPCR

Click here for additional data file.

Additional Information and Declarations

Competing Interests

Author Contributions

Data Availability

The authors declare there are no competing interests.

Ming Li conceived and designed the experiments, performed the experiments, prepared figures and/or tables, authored or reviewed drafts of the paper, and approved the final draft.

Shengli Xia performed the experiments, analyzed the data, authored or reviewed drafts of the paper, and approved the final draft.

Ping Shi performed the experiments, authored or reviewed drafts of the paper, and approved the final draft.

The following information was supplied regarding data availability:

Raw data are available at TCGA (search term: TCGA LIHC), NCBI GEO (GSE114783, GSE128726 and GSE89632), and in the Supplemental Files.

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
