# Peer review of "DPM1 expression as a potential prognostic tumor marker in hepatocellular carcinoma"

_PeerJ, doi:10.7717/peerj.10307_

## Round 0.1 · original submission · Major Revisions

Your manuscript has been reviewed by two experts in the field. As you can see from their comments below, both of them raise substantial points in it. Please read them carefully and revise the manuscript accordingly. Particularly, both of them want more information on the figures. In addition, I feel that more discussion on the relationship between your novel markers and the existing ones should be necessary, as pointed out by Reviewer 1.

Reviewer 1 ·

Basic reporting

no comment.

Experimental design

no comment.

Validity of the findings

no comment.

Additional comments

Major comments:
1. Because HCC usually develops from chronic injured liver, the cause of
chronic liver disease (CLD), such as viral infection, alcohol, fatty
liver, is of importance. They should analyze the relationship between
the expression of DPM1/2/3 and the etiology for CLD.

2. There is no comparative analysis between DPM1/2/3 and already-existed
markers such as AFP, DCP, and GPC3. To highlight the significance of
DPM1/2/3, additional analyses would be necessary.

3. The authors documented that HCC cases with low DPM1/2/3 expression
showed significantly longer overall survival. From a perspective of its
biological function (various protein glycosilation), please discuss why
the DPM1/2/3 overexpression contributes to favorable prognosis.

4. It is hard to realize Fig.5. There is no information about genetic
mutations. Amplification is one of the genetic alteration, but not
mutation.

5. They conducted functional enrichment analyses. However, they analyzed
50 genes similar to DPM1/2/3, but not DPM1/2/3. Please clarify this
issue.

Minor comments:
1. Please spell out all abbreviations when they were firstly written.
Additionally, names of cancer types in Fig.1 were spelled out in its
legend.

2. Please confirm that there is no problem to use images downloaded from
Human Protein Atlas in your manuscript. The reviewer has never
experienced such a case.

3. Labels in Fig. 3 was too unclear to recognize. These should be
modified using drawing software.

Reviewer 2 ·

Basic reporting

This paper presents the results of gene expression analysis of DPM1, DPM2, and DPM3 as a potential prognostic biomarker of hepatocellular cancer patients. It describes interesting data on expression of these genes in retrospective data sets. This research contains interesting findings with a focus on DPMS, but there are several questions and comments.
1. The authors represent the relationship between the DPMS expression profile and survival based on online database. UALCAN, GEPIA, and cBioportal are analytical tools based on TCGA database and others. However, cBioportal uses data of gene amplification as well as gene expression. Although results by each tool were similar, the role of gene amplification could be discussed in cBioportal.
2. Please provide sufficient information in Figure 1. Do the blue and red labels indicate the normal and cancerous samples, respectively?
3. In Figure 1, statistical significance of the difference between normal and cancerous samples should be stated.
4. Please provide sufficient information in Figure 3. What meaning did the colors refer to?
5. Genetic alteration should be replaced to genomic alteration in Figure 5.

Experimental design

no comment

Validity of the findings

no comment

Additional comments

I have no anything else.

---

## Round 0.2 · Major Revisions

Your revised manuscript has been reviewed by the same two reviewers. As you can see from their comments below, one of them now is satisfied with your revision while the other recommends its rejection because no further analyses have been done in response to the reviewer's request. I understand that this work is a computational analysis based on public databases and thus it is difficult to fully meet the reviewer's request in general. However, I wonder if the authors could check, for example, the behaviors of the existing markers in your results and perhaps add one or two figures/tables supporting your results/discussion. Please think about any possibilities that you can do to answer the requests from the reviewer with your best effort.

Reviewer 1 ·

Basic reporting

no comment

Experimental design

no comment

Validity of the findings

no comment

Additional comments

The reviewer considered that major comments 1 (analysis between the relationship between the expression of DPM1/2/3 and the etiology for CLD) and 2 (analysis the relationship between DPM1/2/3 and already-existed markers such as AFP, DCP, and GPC3) was extremely important to improve the quality of your paper. Therefore, additional data based on further analyses, but not discussion, was requested. In addition, they could not respond a major comment 4 (improvement of Fig.5). Alteration of DPM1/2/3 genes would be divided into (Amplification, deletion, somatic mutation etc).
Regrettably, the authors have not followed the reviewers' comments, and the revision is quite insufficient.

Reviewer 2 ·

Basic reporting

no comment

Experimental design

no comment

Validity of the findings

no comment

Additional comments

The manuscript has been revised well, and the authors respond to the comment point by point.

---

## Round 0.3 · accepted · Accept

Your revised manuscript has been reviewed by the original reviewer who gave comments to your previous manuscript. As you can see from the comments below, now the reviewer admits that the revision has been appropriately done. Thus, I am happy to recommend its acceptance. Congratulations!

Reviewer 1 ·

Basic reporting

no comment

Experimental design

no comment

Validity of the findings

no comment

Additional comments

The manuscript was satisfactorily revised compared with the previous version.